# Dynamics Analysis and Deep Learning-Based Fault Diagnosis of Defective Rolling Element Bearing on the Multi-Joint Robot

**Wentao Zhang** **, Ting Zhang *****, Guohua Cui and Ying Pan**

School of Mechanical and Automotive Engineering, Shanghai University of Engineering Science, Shanghai 201620, China
* Correspondence: zhangt@sues.edu.cn; Tel.: +86-188-0196-9968

**Abstract:** Industrial robots typically perform a variety of tasks and occupy critical positions in modern manufacturing fields. When certain failures occur in the internal structures of robots, it tends to result in significant financial loss and the consumption of human resources. As a result, timely and effective fault diagnosis is critical to ensuring the safe operation of robots. Deep learning-based methods are currently being widely used by researchers to address some shortcomings of traditional methods. However, due to realistic factor limitations, few researchers take the failure pattern of rotating machinery and the location of fault joints into account at the same time, so the fault types of multi-joint robots are not thoroughly investigated. In this case, we proposed a dynamic simulation method that considers the effects of bearing failures at various faulty joint locations and makes it possible to investigate more possible failure cases of multi-joint robots. In addition, we used LSTM and DCNN to perform staged fault diagnosis tasks, allowing us to gradually locate faulty joints and investigate detailed failure forms. According to the experimental results, distinguishable vibration signals corresponding to various fault states are successfully obtained, and our implemented algorithms are validated for their advanced performance in diagnosis tasks via comparative experiments.

**Keywords:** robot dynamics; fault diagnosis; rolling element bearing; deep learning; long short-term memory; deep convolutional neural network



## 1. Introduction

Industrial robots have been widely used in the manufacturing system of the current production process due to their high levels of efficiency, accuracy, and flexibility, and carry out numerous critical tasks such as welding [1], polishing [2], assembling [3], spraying [4], carrying [5], and so on. When certain industrial robots malfunction, it often results in a stalled production line, the consumption of human and material resources, and even the personal safety of employees. The rotating joints, as the main component of the mechanical transmission system, are the critical structure for transferring motion and force for a multi-joint robot. Rolling element bearings are the most likely to lose efficacy among the rotating machinery included in the joints due to volatile loads and speeds. As a result, it is critical to implement timely and effective fault diagnosis to monitor the running state of industrial robots, particularly the health state of bearings located in rotating joints, in order to ensure the long-term safe and reliable operation of mechanical equipment.

At present, the issue of fault diagnosis has attracted widespread attention from researchers in engineering application fields, and more diagnostic methods have been proposed. Wu et al. [6] proposed a dynamic model for squirrel-caged induction to analyze broken-rotor-bar and turn-to-turn short faults. Song et al. [7] proposed a new heavy tail degradation model for predicting rolling bearings' useful life. Furthermore, machine learning is a technology that can automatically learn features from collected data and create an intelligent prediction model. Jaber et al. [8] proposed a method for detecting gearbox faults in the PUMA 560 robot that uses a discrete wavelet transform (DWT) to extract time-frequency features and an artificial neural network (ANN) to realize fault classification.

Hsu et al. [9] used multi-class support vector machines (SVMs) in conjunction with principal component analysis (PCA) to diagnose several real aging-related faults on a six-axis robot, and the experimental results validated their findings effectively. Lu et al. [10] designed an enhanced k-nearest neighbor (KNN) embedded with a sparse filtering extractor that can select the health state label of rotating machinery adaptively based on the optimized correlation vectors. Fang et al. [11] used data dimension reduction and random forest (RF) to detect loose screw faults in the SCARA robot, and the applied design shows a good performance. However, the above-mentioned methods have limitations in complicated feature extraction and insufficient generalization, so deep learning (DL)-based methods are proposed to address these shortcomings.

DL is an important branch of machine learning that has expanded the field of artificial intelligence applications. Because of its end-to-end characteristics and adaptive feature extraction abilities, DL can significantly reduce reliance on human intervention and has numerous successful application cases in the fault diagnosis field [12]. Jiao et al. [13] used an improved D-S evidence theory and a deep belief network (DBN) to predict bearing failure on the industrial robot's joint, eventually achieving an average accuracy of about 98%. Pan et al. [14] developed a deep convolutional neural network (DCNN)-based fused sensor and actuator fault diagnosis model for the robot joint and demonstrated its effectiveness by achieving high fault recognition accuracy. Hong et al. [15] developed an attitude data-based intelligent fault identification approach by training a deep sparse auto-encoder network (DSAE), which has effective performance for multi-joint robot fault identification. Xia et al. [16] proposed a novel deep perceptual adversarial domain adaptive (DPADA) method for fault diagnosis of robot bearings under varying conditions, which outperforms convolutional neural network (CNN) and conditional domain-adversarial network (CDAN)-based methods.

Even though these DL-based methods have increased the practicability and generalizability of the algorithm, there are still several issues and deficiencies that have not been addressed. It has been discovered that researchers tend to investigate failures on a few specific joints without considering the state of others in a multi-joint robot, resulting in insufficient attention to the overall system's running state. Furthermore, few works simultaneously take the failure pattern of rotating machinery and the location of fault joints into account so that the amount of identifiable failure patterns is limited. It is difficult to collect characteristic signals corresponding to the fault state under more complicated conditions due to the limitations of some realistic factors. Therefore, we proposed a novel dynamic modeling method for multi-joint robots that can simulate rotating machinery faults on different robot joints and allow us to investigate more types of faulty operating states. Furthermore, we used DL-based methods to complete the diagnosis tasks in stages and achieved high accuracies to validate the superior performance of the used models. In summary, the following are the main contributions of this work:

1.  The failure mechanism of rotating machinery is investigated, with the rolling element bearing serving as the primary research object. The effects of bearing failures corresponding to different faulty joint locations are considered in the multi-joint robot's dynamic simulation, so that more possible failure modes can be investigated by collecting vibration signals from all joints.

2.  We developed a staged workflow for detecting bearing failures in multi-joint robots. The long short-term memory (LSTM) network is introduced in the first stage to recognize the health status of each joint and lock the positions of faulty joints. In the second stage, the signals of the faulty joint are extracted separately, time-frequency imaging is implemented, and then DCNN is used to identify the detailed failure form of the roller element bearings of the faulty joints. This allows for the determination of both fault locations and causes.

3.  In addition to LSTM and DCNN, other fault diagnosis methods such as back-propagation neural network (BPNN), support vector machine (SVM), and naive Bayes classifier (NBC)

are tested and used as comparing algorithms to provide a comprehensive performance evaluation.

The remainder of this paper can be summarized as follows: Section 2 introduces the research object and methods used in this work, such as the kinetic and dynamic analysis of the multi-joint robot, the fundamental theories of the deep learning algorithms used, LSTM and CNN, and the general procedure of the research workflow in this work. Section 3 explains the rolling bearing fault principle and demonstrates the dynamic simulation process of a multi-joint robot with bearing failure. Section 4 focuses on the experimental results obtained, including data visualization, the introduction of some experimental details, algorithm performance validation, and result comparisons. Section 5 presents the conclusions and future work.

## 2. Research Object and Methods

### 2.1. Brief of the Multi-Joint Robot

The multi-joint robot is essentially a mechanical arm with numerous connecting rods and joints. The 6-axis robot has a broader range of applications in industrial production due to its greater movement flexibility. In this study, a 6-axis multi-joint industrial robot named "Gluon-6L3" is used as an analysis object, and the real product and simulation model figures are shown in Figure 1, where the lower three joints primarily determine the position of the end executor and the upper three joints affect the attitude. The driving and transmission mechanisms in the robot are prone to failure during operation due to the effect of load and torque, which can be expressed as changes in motion attitude. Hence, it is worthwhile to figure out the kinematic and dynamic characteristics of the multi-joint robot to analyze the fault principle of defective elements.

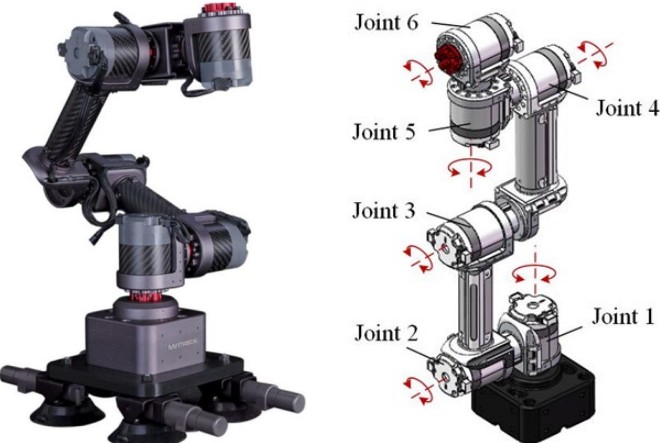

**Figure 1.** The figure of the real product and the simulation model of the multi-joint robot.

To analyze the displacement and velocity relationship between the links of the manipulator, the Denavit-Hartenberg (D-H) parameter method [17] is adopted to describe the relative translation and rotation of joints. After the establishment of a localized coordinate system, the relationship and transformation mechanism of any adjacent links can be obtained and expressed as the following homogeneous matrix:

$$
{}^{i-1}_{i}T = \begin{bmatrix} c\theta_i & -c\alpha_i s\theta_i & s\alpha_i s\theta_i & a_i s\theta_i \\ s\theta_i & c\alpha_i c\theta_i & -s\alpha_i c\theta_i & a_i s\theta_i \\ 0 & s\alpha_i & c\alpha_i & d_i \\ 0 & 0 & 0 & 1 \end{bmatrix} \tag{1}
$$

where ${}^{i-1}_{i}T$ represents the transfer matrix from the $i-1st$ frame to the $i$th frame, four parameters $a_i$, $\alpha_i$, $d_i$, and $\theta_i$, denote the rod length, torsional angle, offset distance, and joint angle, respectively. $c\theta$ and $s\theta$ are the shorthand form of $\cos\theta$ and $\sin\theta$, respectively.

The local coordinate systems are established on the motor centers of joints, and then the rod parameters are measured for kinetic modeling, which is shown in Table 1.

**Table 1.** The rod parameters of the 6R-robot.

| Rod | $\theta_i/\degree$ | $\alpha_i/\degree$ | $a_i$/mm | $d_i$/mm | Range of $\theta_i/\degree$ |
|-----|------|------|------|------|------|
| 1 | $\theta_1$ | 90 | 0 | 158.5 | $-140{\sim}140$ |
| 2 | $\theta_2$ | 0 | 173 | 0 | $-90{\sim}90$ |
| 3 | $\theta_3$ | 0 | 173 | 0 | $-140{\sim}140$ |
| 4 | $\theta_4$ | 90 | 0 | 79.2 | $-140{\sim}140$ |
| 5 | $\theta_5$ | $-90$ | 0 | 79.2 | $-140{\sim}140$ |
| 6 | $\theta_6$ | 0 | 0 | 41.7 | $-360{\sim}360$ |

Furthermore, the relationship between joint torques and motions of the multi-joint robot can be described as the following standard form:

$$\tau = M(q)\ddot{q} + C(q,\dot{q}) + G(q) \tag{2}$$

where $\tau$ is the generalized moment vectors on each joint, and $q$, $\dot{q}$, and $\ddot{q}$ are the vectors of joint displacement, velocity, and acceleration, respectively. $M(q)$ is the inertia matrix, $C(q,\dot{q})$ is the centrifugal and Coriolis vector, and $G(q)$ is the gravity vector [18]. By implementing the abovementioned dynamic equations, the required driving torques of each joint can be calculated according to the planned motion.

### 2.2. Deep Learning-Based Algorithm
### 2.2.1. Long Short-Term Memory Network

LSTM is a deep learning algorithm that is commonly used to deal with time series learning problems. The LSTM introduces a gating mechanism to control the forgetting and updating of information. Several internal functions, such as the forgetting gate, the input gate, and the output gate, can work together to regulate the information flow inside the cell and make the LSTM selectively accept information from previous time steps. Figure 2 depicts the basic components of the LSTM network's hidden node structure [19].

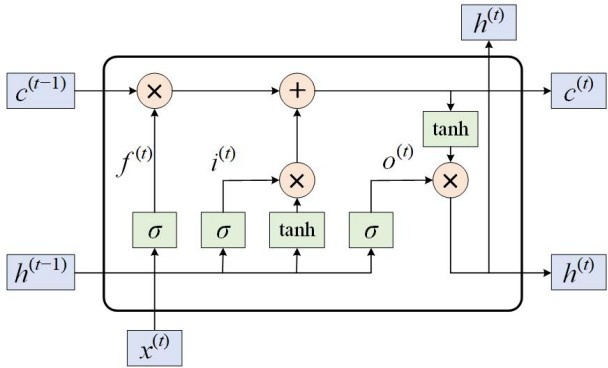

**Figure 2.** The basic structure of LSTM.

In Figure 2, the parameters $x^{(t)}$, $c^{(t)}$, and $h^{(t)}$ represent the input vector, state vector, and output vector, respectively. $h^{(t)}$ at time step $t$ is the fusion of $x_t$, forgetting gate $f^{(t)}$, input gate $i^{(t)}$, output gate $o^{(t)}$, memory cell $\tilde{c}^{(t)}$, and $h^{(t-1)}$ from the last time step. The calculating formulas are expressed as follows:

$$f^{(t)} = \sigma\left(W_f\left[h^{(t-1)}, x^{(t)}\right] + b_f\right) \tag{3}$$

$$i^{(t)} = \sigma\left(W_i\left[h^{(t-1)}, x^{(t)}\right] + b_i\right) \tag{4}$$

$$o^{(t)} = \sigma\left(W_o\left[h^{(t-1)}, x^{(t)}\right] + b_o\right) \tag{5}$$

$$\widetilde{c}^{(t)} = \tan h\left(W_c\left[h^{(t-1)}, x^{(t)}\right] + b_c\right) \tag{6}$$

$$c^{(t)} = i^{(t)}\widetilde{c}^{(t)} + f^{(t)}c^{(t-1)} \tag{7}$$

$$h^{(t)} = o^{(t)} \cdot \tan h\left(c^{(t)}\right) \tag{8}$$

where $W_f$, $W_i$, $W_o$, and $W_c$ are weight matrixes of $f^{(t)}$, $i^{(t)}$, $o^{(t)}$, and $\widetilde{c}^{(t)}$, respectively; $b_f$, $b_i$, $b_o$, and $b_c$ are corresponding bias terms; and $\sigma$ is the sigmoid function.

### 2.2.2. Convolutional Neural Network

CNN is a commonly used deep learning algorithm in image processing. Because of the characteristics of local receptive fields, weight sharing, and sparse connections, it typically has fewer parameters than a fully connected network [20]. As shown in Figure 3, the classical CNN structure consists of convolutional layers, pooling layers, and fully connected layers.

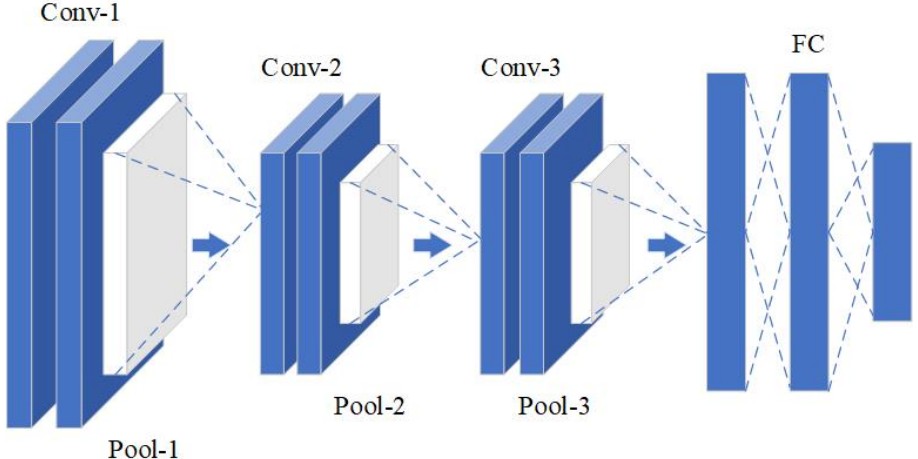

**Figure 3.** The basic structure of CNN.

The convolutional layers are comprised of convolution filters with multiple kernels, and they extract abstract features from input graphs. The pooling layers can reduce the redundant information of feature data and improve the calculation efficiency of the algorithm. Their mathematical calculations are as follows:

$$x_j^l = f\left(\sum_{i \in M_j} x_i^{l-1} * k_{ij}^l + b_j^l\right) \tag{9}$$

$$x_j^l = f\left(\beta_j^l down\left(x_i^{l-1}\right) + b_j^l\right) \tag{10}$$

where $M_j$ represents the localized receptive area; $k_{ij}^l$ and $b_j^l$ are the weight and bias value of the $l$th layer, respectively. $x_i^{l-1}$ is the pixel value of feature images, operator * represents convolution operation, $f(\cdot)$ is the activation function, and $down(\cdot)$ indicates the subsampling operation of maximum pooling. After the multiple convolutional and pooling operations, the fully connected layers receive the extracted features and output predicted results as a classifier.

### 2.3. General Procedure of Diagnosis Process

To achieve the accurate fault diagnosis of rotating machinery failure in joints, the vibration signals are needed to reflect the running state of the multi-joint robot. On account of the difficulties to acquire data corresponding to the failure state in reality, the virtual

prototype of a 6-axis multi-joint industrial robot was built in the simulation environment to explore the influence of joint failure.

The related parameters used for dynamics modeling are measured in Solidworks, and the required driving torques of each joint are calculated by the robotics toolbox [21] in MATLAB. By importing driving torques and adding fault excitation into robot joints, the running state of the multi-joint robot with bearing failure occurring in the joint can be simulated through the dynamic analysis in ADAMS. The vibration signals collected by acceleration sensors are selected as the feature variables to be used in fault diagnosis.

The fault diagnosis of the multi-joint robot is divided into two stages. The first stage is implementing LSTM to investigate the whole state of the robot. The collected vibration signals of all joints are merged as the multi-channel time series to build a fault dataset for training LSTM so that it can recognize the health state of all joints at once. If there is an anomaly discovered, like some abnormal shakes or noises, LSTM should preliminarily lock the approximate scope by pointing out which joint is anomalous.

Then, in the second stage, DCNN will be adopted to separately diagnose the detailed fault pattern of this faulty joint, such as a scratch or wear occurring on the surface of the inner race or outer race of the motor bearing in this robot joint. DCNN can only recognize the state of one joint due to its function, and the form of its input data are two-dimensional images transformed from single-channel vibration signals of the faulty joint through continuous wavelet transform (CWT). In this way, the staged fault diagnosis process can be achieved from whole to part, and the workflow chart of the abovementioned procedure is summarized in Figure 4.

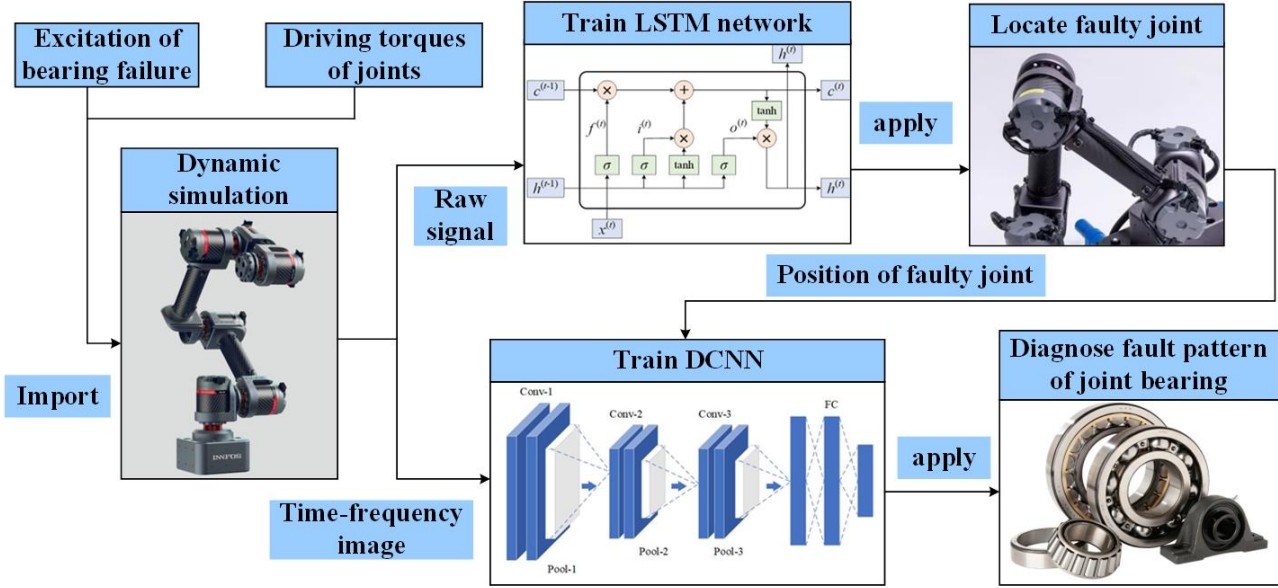

**Figure 4.** The workflow chart of fault diagnosis of multi-joint robot.

## 3. Dynamic Simulation of Fault States

### 3.1. Contact Deformation of Bearing Failure

The bearing failure accounts for about 40% of mechanical failures according to the survey, and over 90% of the faults in rolling bearings occur in the inner and outer raceway [22]. Therefore, the types of bearing faults analyzed in this work are mainly single-point faults occurring in the inner and outer rings of bearings.

It is assumed that the inner race keeps rotating at a constant speed and the outer ring is stationary under the external radial load. Each ball performs pure rolling without slip, and the effect of weight is ignored. The scratch defect model of the raceway surface of the inner and outer race of the bearing is shown in Figure 5.

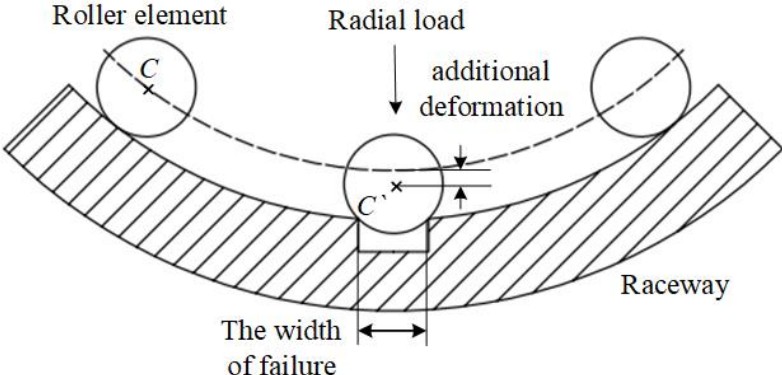

**Figure 5.** The scratch defect model of rolling bearing.

At the early stage of the scratch formation, the defect is usually localized so that the roller elements will pass through the defect area without touching the bottom [23]. When the roller element is in contact with the damaged point on the raceway surface, an additional elastic deformation will be generated. Due to the Hertz theory, the additional contact force $P$ caused by the defect is calculated as follows [24]:

$$P = \pi \kappa E \sqrt{\frac{2\varepsilon R \delta^3}{9F^3}} \tag{11}$$

$$\delta = \frac{d_b}{2}\left(1 - \cos\frac{w_b}{d}\right) \pm \frac{d}{2}\left(1 - \cos\frac{w_d}{d_b}\right) \tag{12}$$

where $\delta$ is the additional contact deformation caused by failure; $E$ is the parameter determined by the elastic modulus and Poisson ratio; $R$ is the effective radius of curve; $\varepsilon$ and $F$ are the complete elliptic integral of the first and second kind, respectively; $d$ denotes the diameter of the raceway where the defect is located; $w_d$ represents the width of the failure; and "+" and "−" correspond to the scratch located in the inner and outer race, respectively.

### 3.2. Excitation Signals of Bearing Failure

#### 3.2.1. Outer Race Fault

As the roller elements keep rotating, a series of impacts will occur at a certain frequency, and different fault locations correspond to different characteristic frequencies. When a fault occurs in the outer ring of the bearing, the magnitude and direction of the impulse force caused by the fault point will not change because of the fixed position of the outer ring. A series of impacts generated by outer race fault (OF) can be expressed as [25]:

$$\Delta_o(t) = \sum_{k=1}^{\infty} P_o \delta\left(t - \frac{k}{f_o}\right) \tag{13}$$

$$f_o = \frac{Z}{2} f_r \left(1 - \frac{d_b}{D_m}\cos\alpha\right) \tag{14}$$

where $P_o$ is additional contact force caused by OF; $\delta(t)$ represents the impulse function occurring when $t = 0$; $f_o$ is the OF characteristic frequencies and determines the interval between the two adjacent impacts; $k$ represents the positive integer number; $f_r$ is the shaft frequency; $Z$ is the number of rolling elements; $\alpha$ is the contact angle, for a deep groove bearing $\alpha = 0$; $D_m$ is the pitch diameter; and $d_b$ is the diameter of a rolling element. The OF excitation signal and response signal are shown as Figure 6.

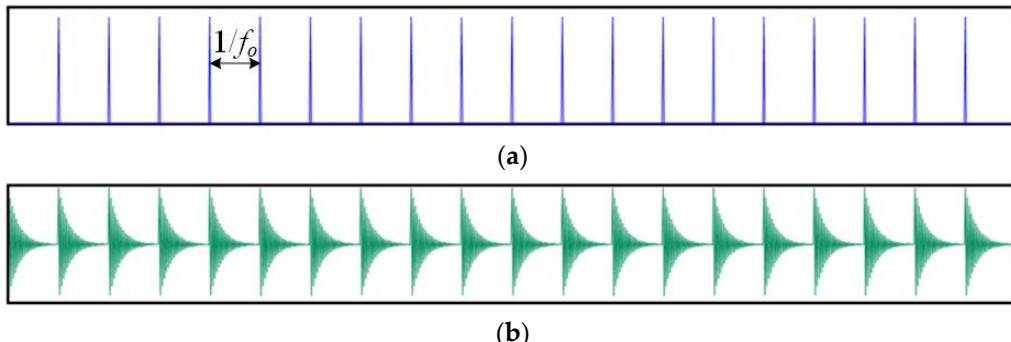

**Figure 6.** OF Excitation signal and response signal. (**a**) The excitation signal of OF; (**b**) The response signal of OF impact.

### 3.2.2. Inner Race Fault

When the fault point is located in the inner ring, the fault point will move with the continuously rotating inner race, and the sensor is usually fixed on the shell that remains stationary. Therefore, when the fault point and the sensor orientation have an angle, the sensor can only receive the component force of the pulse force, and a series of impacts generated by inner race fault (IF) can be expressed as:

$$\Delta_i(t) = \sum_{k=1}^{\infty} P_i \delta\left(t - \frac{k}{f_i}\right) |\cos(2\pi f_r t)| \tag{15}$$

$$f_i = \frac{Z}{2} f_r \left(1 + \frac{d_b}{D_m} \cos\alpha\right) \tag{16}$$

where $P_i$ is additional contact force caused by IF, and $f_o$ is the IF characteristic frequencies. The IF excitation signal and response signal are shown as Figure 7.

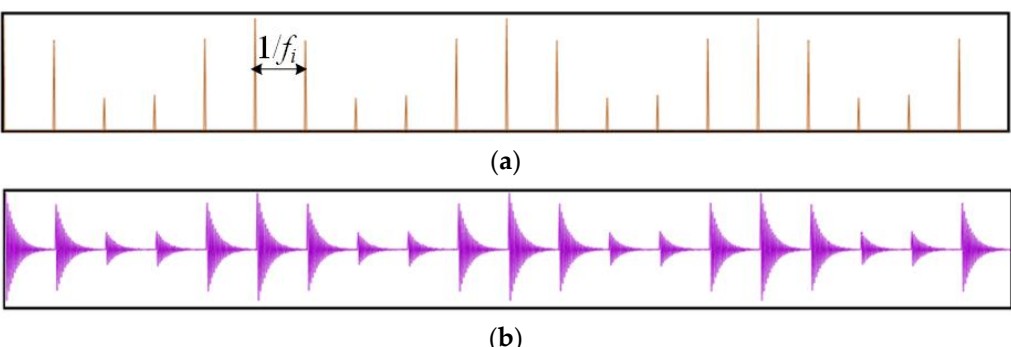

**Figure 7.** IF Excitation signal and response signal. (**a**) The excitation signal of IF; (**b**) The response signal of IF impact.

The geometrical parameters of the deep groove ball bearing, whose designation is 6205-2RS, are taken to obtain the characteristic frequencies of bearings, where $D_m = 39.04$ mm, $d_b = 7.94$ mm, and $Z = 9$. Additionally, the material attributes of bearing steel GCr15 [26] are used to calculate the additional contact load caused by failures, where the elastic modulus is $207 GPa$ and the Poisson ratio is 0.29. Therefore, when $w_d = 0.5$ mm and $f_r = 30$ Hz, the OF characteristic frequency and contact force are $f_o = 107.54$ Hz, $P_o = 0.4311$ N, respectively, and for IF, $f_i = 162.46$ Hz, $P_i = 0.7950$ N.

### 3.3. Simulation of Joint with Bearing Failure

The fault states of the robot joint are simulated through the dynamics analysis in the ADAMS environment. First of all, the driving torques of all joints and fault excitation signals of bearing outer and inner race faults are imported as spline functions to the data

units, and they will be used as the applied load to affect the running states of joints. The process of applying torques and excitations is achieved by setting general forces in the robot joints.

The general force is a type of force vector with multi-components, including X-force, Y-force, and Z-force as the 3 component forces, and X-torque, Y-force, and Z-force as the 3 component torques, as shown in Figure 8. Due to the X-axis being prescribed in the same direction as rotation axis of joint, the X-torque is assigned the value of driving torque by the AKISPL function. When this joint is healthy, other components of general force should be zero.

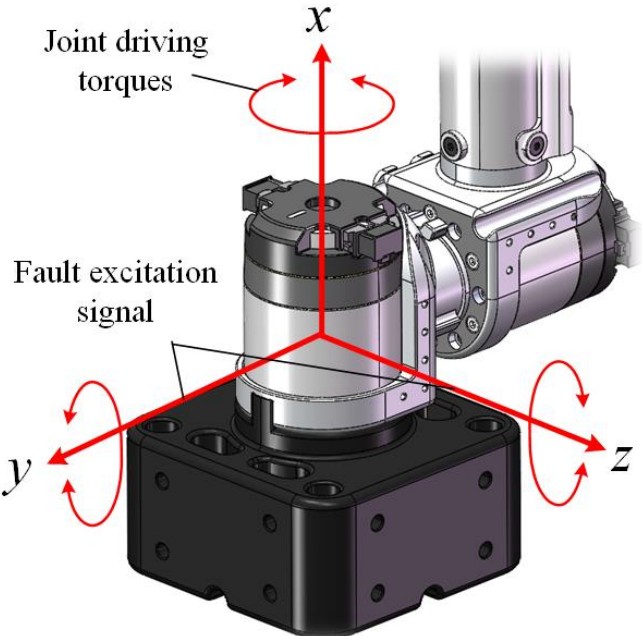

**Figure 8.** Diagram of the general force set in robot joint.

If the fault state of this joint needs to be investigated, for example, if the bearing inside has an OF, the periodical impacts will occur in the radial direction of the bearing which is perpendicular to the X-axis, so the Y-force or Z-force can be assigned the value of the excitation signal of the outer race fault. In fact, any vector through the origin in the *yoz* plane can be taken as the direction of excitation signals, and that is determined by the orientation of the fault point in the outer raceway. The case of IF is a similar case. However, the direction of excitation will continuously change due to the rotation of the inner race, so the inner race fault should be synthesized by two variational components (Excitation_C1 and Excitation_C2). The set principles of the joint state are listed in Table 2.

**Table 2.** The set principle of the joint health state.

| Component | Health State | OF in Y-Axis | OF in Z-Axis | IF |
|---|---|---|---|---|
| X-force | 0 | 0 | 0 | 0 |
| Y-force | 0 | OF Excitation | 0 | IF Excitation_C1 |
| Z-force | 0 | 0 | OF Excitation | IF Excitation_C2 |
| X-torque | Driving Torque | Driving Torque | Driving Torque | Driving Torque |
| Y-torque | 0 | 0 | 0 | 0 |
| Z-torque | 0 | 0 | 0 | 0 |

By exerting different impacts on different joints, the running statues with different fault locations or different fault types are simulated. In this way, the data can be acquired corresponding to different labels, which will be predicted by the diagnosis algorithms.

### 3.4. Vibrational Data Acquirement

To achieve the movement of the robot, the driving torques of the joints are calculated according to the planned motion. The motion is comprised of the angular displacements of six joints, and there are two different groups of motions taken as analysis examples, which are substituted into dynamics equations to solve the corresponding joint torques. It is assumed that the robot in realistic industrial applications tends to work with a group of planned actions repeatedly, and the length of their motion cycle is assumed to be 10 s. In this way, the curves of applied motions of one cycle are shown in Figure 9.

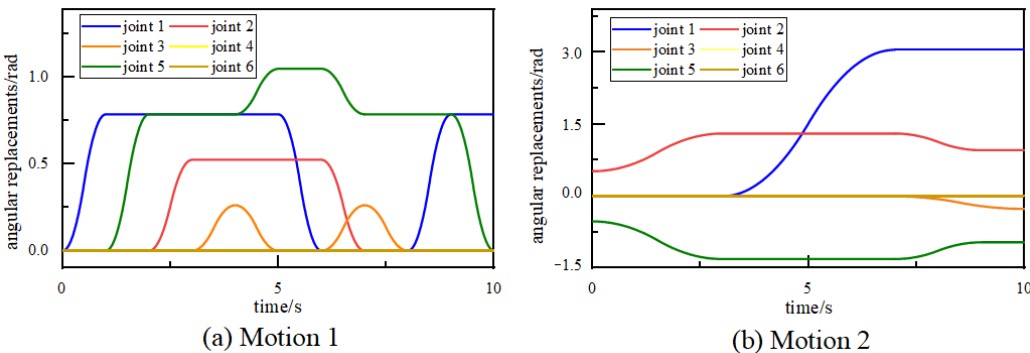

**Figure 9.** The curves of angular displacements applied in this work. (**a**) Motion 1; (**b**) Motion 2.

The sample interval is 0.01 s so that one second includes 100 data points finally. The acceleration data measured on the outer sphere of the six joints are collected to establish the six-channel datasets, which represents the state of the whole robot system.

Several collected vibration signals in joint 2 with different running states are shown in Figure 10. It can be observed that the different fault locations have different impacts on the running state of the robot. Because the generation of failure is related to the motion state of each joint, the fault impacts from various locations usually occur when the corresponding joint is moving, and are accompanied by apparent trembles.

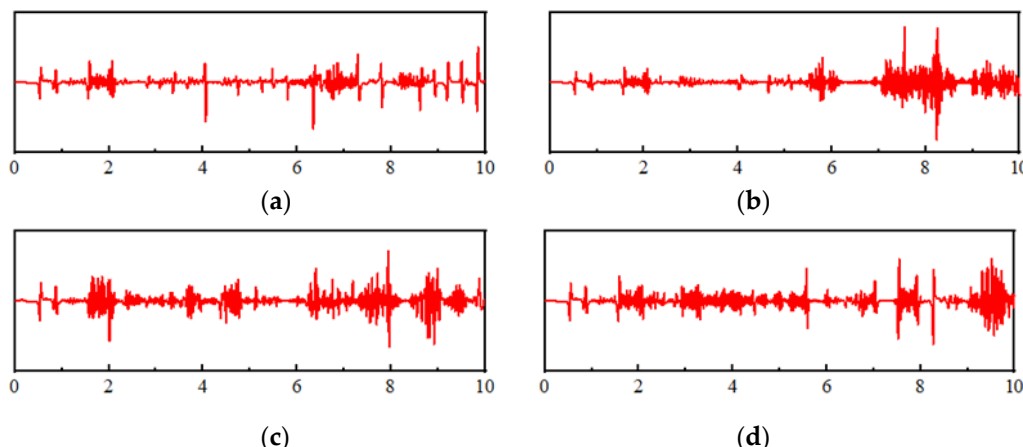

**Figure 10.** The collected vibration signals in joint 2 with various states. (**a**) Health state; (**b**) Failure in joint 2; (**c**) Failure in joint 3; (**d**) Failure in joint 4.

## 4. Fault Diagnosis of Multi-Joint Robot

### 4.1. Diagnosis of Faulty Joint Location Based on LSTM

#### 4.1.1. Description of Fault Datasets

The section is the first stage of the diagnosis procedure, and the LSTM network is adopted to figure out which joint is at fault. To evaluate the performance of the algorithm more comprehensively, the collected vibrational signals are assembled as single-joint failure (SF) datasets and multi-joint failure (MF) datasets. Each of them includes a health state and

six types of fault states, for a total of seven states marked as label number 0~7, where label 0 represents that all joints are healthy. Each failure state contains 1000 samples in the training set and 200 samples in the testing dataset after data augmentation. The detailed information about SF and MF datasets is listed in Table 3. This table shows the health states of each joint of different label, where "0" represents this joint is health and "1" means this joint has a failure. Moreover, these datasets are further divided into two parts corresponding to two different groups of motion so that the influence of different planned actions can be observed.

**Table 3.** The overview of SF and MF dataset.

| Dataset | Label | Health State of Each Joint (0: Healthy, 1: Faulty) | | | | | |
| | | Joint 1 | Joint 2 | Joint 3 | Joint 4 | Joint 5 | Joint 6 |
| --- | --- | --- | --- | --- | --- | --- | --- |
| SF | SF1 | 1 | 0 | 0 | 0 | 0 | 0 |
| | SF2 | 0 | 1 | 0 | 0 | 0 | 0 |
| | SF3 | 0 | 0 | 1 | 0 | 0 | 0 |
| | SF4 | 0 | 0 | 0 | 1 | 0 | 0 |
| | SF5 | 0 | 0 | 0 | 0 | 1 | 0 |
| | SF6 | 0 | 0 | 0 | 0 | 0 | 1 |
| MF | MF1 | 1 | 1 | 0 | 0 | 0 | 0 |
| | MF2 | 0 | 1 | 1 | 0 | 0 | 0 |
| | MF3 | 1 | 1 | 1 | 0 | 0 | 0 |
| | MF4 | 0 | 1 | 1 | 1 | 0 | 0 |
| | MF5 | 1 | 1 | 1 | 1 | 0 | 0 |
| | MF6 | 1 | 1 | 1 | 1 | 1 | 0 |

Each sample is an array constituted by multiple one-dimensional time series with six channels, where each data channel corresponds to the vibration signal of a joint. Each sequence is composed of 1000 data points intercepted in the time period, so each sample is a two-dimensional array of size (1000, 6), where the first dimension is the sequence length corresponding to 1000 data points in the time dimension, and the second dimension is the feature vector length corresponding to six data channels.

4.1.2. Related Settings of LSTM

The LSTM network is built by Keras, an open-source neural network computing library in the deep learning framework Tensorflow. The input dimension of the LSTM layer is set to (None, 1000, 6), the memory vector length is 500, and the output of the last time step is returned by default. The LSTM layer is regarded as a feature extractor, so it needs to be followed by a dense layer as a classifier. The number of the neurons in the dense layer is set to 7, which is the same as the number of label types contained in the dataset. The activation function softmax is used in the multi-classification problem to return a set of probability distributions predicting the probability that a sample belongs to each label category. The finally formed LSTM network is the architecture consisting of two layers and 1,007,006 trainable parameters.

The layers of completed LSTM network is encapsulated into an overall network model through the network container "Sequential" provided by Keras. Only the instance of the network model needs to be called to complete the sequential propagation calculation of data from the first layer to the last layer. The "model.pile()" function is applied to specify the optimizer, learning rate, error function and evaluation index used by the network, and then send the training dataset to the network and set the number of epochs through the "model.fit()" function so that the network model can start to be trained. Table 4 summarizes the relevant parameter settings for training LSTM network.

**Table 4.** The relevant parameter settings for LSTM network.

| Parameter Item | Value |
|---|---|
| Learning rate | 0.01 |
| Loss function | Cross-entropy loss |
| Optimizer | Adam |
| Epoch | 20 |
| Batch size | 50 |
| Validation ratio | 0.1 |

4.1.3. Diagnosis Results of Comparative Experiments

Table 5 shows the recognition results obtained by LSTM on SF and MF datasets, which involve several evaluation criteria like precision, recall, and F1-score. Precision is the ratio of truly predicted positive samples in all predicted positive results, recall is the ratio of truly predicted positive samples to all actual positive samples, and F1 is the comprehensive consideration of them. From these indexes, the details of the misjudgment of the algorithm can be learned. For example, for the MF dataset of "Motion 2", the recall of "MF3" and the precision of "MF5" are relatively low, so the LSTM is likely to misjudge the "MF3" samples as "MF5" samples. Beyond that, the possibilities of the misjudgment of other situations are very low.

**Table 5.** The recognition results of LSTM on SF and MF dataset.

| Dataset | Label | Motion 1 | | | Motion 2 | | |
|---|---|---|---|---|---|---|---|
| | | Precision | Recall | F1-Score | Precision | Recall | F1-Score |
| SF | Norm | 0.9780 | 1.0000 | 0.9889 | 0.9950 | 0.9950 | 0.9950 |
| | SF1 | 1.0000 | 0.9975 | 0.9987 | 0.9950 | 0.9975 | 0.9963 |
| | SF2 | 0.9895 | 0.9450 | 0.9668 | 0.9852 | 0.9975 | 0.9913 |
| | SF3 | 0.9901 | 0.9975 | 0.9938 | 0.9975 | 1.0000 | 0.9988 |
| | SF4 | 1.0000 | 0.9925 | 0.9962 | 0.9975 | 0.9925 | 0.9950 |
| | SF5 | 0.9513 | 0.9775 | 0.9642 | 1.0000 | 0.9975 | 0.9987 |
| | SF6 | 0.9774 | 0.9750 | 0.9762 | 0.9949 | 0.9850 | 0.9899 |
| MF | Norm | 0.9707 | 0.9950 | 0.9827 | 0.9974 | 0.9525 | 0.9744 |
| | MF1 | 0.9973 | 0.9175 | 0.9557 | 0.9515 | 0.9800 | 0.9655 |
| | MF2 | 0.9875 | 0.9900 | 0.9888 | 0.9779 | 0.9950 | 0.9864 |
| | MF3 | 0.9366 | 0.9975 | 0.9661 | 0.9602 | 0.8450 | 0.8989 |
| | MF4 | 1.0000 | 0.9825 | 0.9912 | 0.9975 | 0.9975 | 0.9975 |
| | MF5 | 1.0000 | 1.0000 | 1.0000 | 0.8753 | 1.0000 | 0.9335 |
| | MF6 | 0.9925 | 0.9975 | 0.9950 | 0.9872 | 0.9625 | 0.9747 |

In order to more comprehensively verify the effectiveness and advancement of the LSTM network, multiple comparison algorithms are set up in the experiment process to carry out the same experiment, and finally the recognition accuracy on the test set is used as an index for comparative evaluation. The comparative algorithms include naive Bayes classifier (NBC) [27], support vector machine (SVM), back-propagation neural network (BPNN), and simple RNN. Among these algorithms, RNN and LSTM can directly take multi-channel time series as input, so there is no additional processing for collected vibration signals except simple data normalization when RNN or LSTM is implemented. However, only one-dimensional arrays can be accepted as input ports in BPNN, SVM, and NBC, so the original signals need data dimension reduction to be conducted when these algorithms are used. The wavelet packets decompose (WPT) [28] is applied to achieve feature extraction by decomposing the signals into several sub-bands with different frequencies and calculating the corresponding energy entropies. In this way, the results can be integrated into low-dimensional vectors as input data space.

The final results are shown in Figure 11, where the scenarios, such as different groups of motions, are illustrated separately. The recognition accuracies are used to evaluate the

capabilities of algorithms. It is clear that LSTM shows the most outstanding performance, achieving the highest accuracy levels across the board, while the simple RNN model performs slightly worse. The experiments may also reveal that different motions have a negligible impact, demonstrating that the diagnosis approach is applicable to the general movements of a multi-joint robot. Based on the results of other algorithms, it is possible to conclude that diagnosis accuracy decreases as the datasets contain more faulty joints, owing to the increased recognition difficulties caused by more complicated interference. The accuracy achieved by BPNN, SVM, and NBC appears to be lower than the first two algorithms, demonstrating that shallow architectures perform poorly when dealing with problems with larger data scales. Finally, even in complex situations, LSTM networks can greatly accelerate fault diagnosis tasks to locate the joint failure of a multi-joint robot.

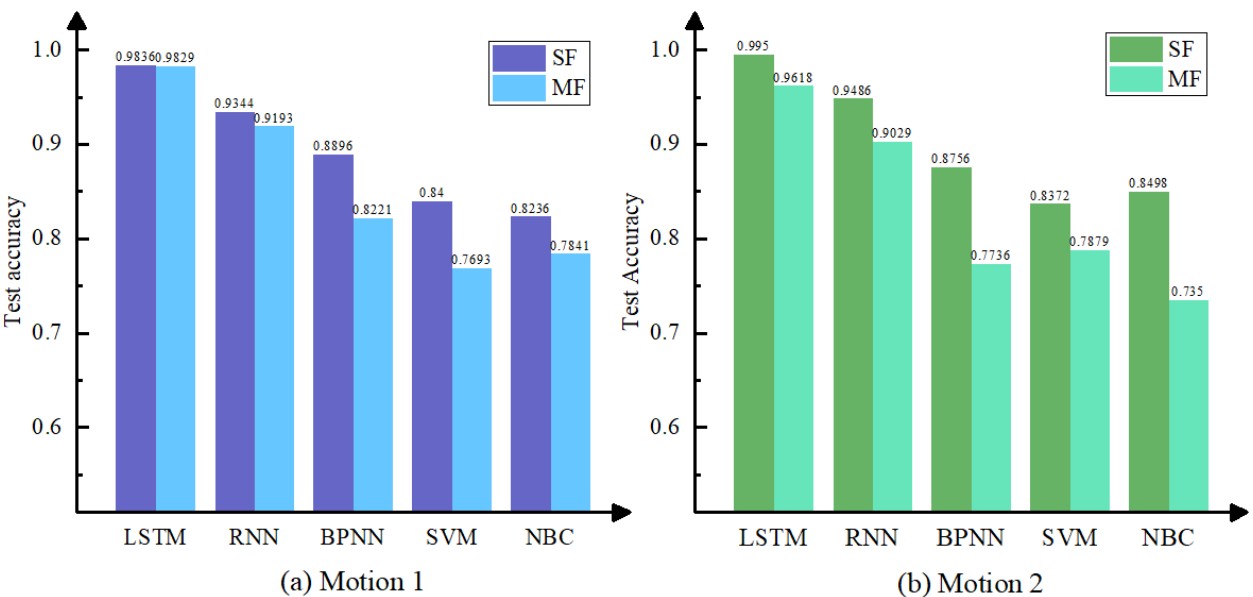

**Figure 11.** Comparisons of recognition accuracies in the first stage. (**a**) Motion 1; (**b**) Motion 2.

### 4.2. Diagnosis of the Joint Failure Type Based on DCNN
#### 4.2.1. Description of Fault Datasets

Following the precise location of the faulty joint, the next step is to determine the causes of the failure in order to complete a more specific diagnosis. The running state of the entire robot is not addressed in this section. Instead, the faulty joint is the primary analysis object, and the single-channel vibration signals from the faulty joint are used as the recognition algorithm's input dataset. According to the fault excitation signals composited in the previous section, we can simulate several fault patterns, including IF and OF, and detailed information about general force settings on different joint fault types (JF) is shown in Table 6. The changes of setting only exist in Y-force and Z-force, so Table 6 only shows information about these two components. Moreover, these datasets are further divided into two parts according to SF and MF: the two different cases, namely the two situations that only this joint has a failure or another joint is also faulty are investigated separately.

In order to obtain sufficient samples, we did manage to collect the vibration signals from faulty joints from more channels and conduct an overlap sampling method. As a result, there are 500 samples in the training set and 100 samples in the testing set for each running state. Subsequently, continuous wavelet transform (CWT) is implemented to transfer one-dimensional vibration signals into time-frequency images which can simultaneously represent the feature information of the time-domain and frequency-domain. Each sample is a matrix with the shape (140, 140, 3), which means each image is 140 × 140 in size and has 3 RGB color channels. In this way, DCNNs, which are applicable to image recognition, are used to conduct fault diagnosis in this stage.

**Table 6.** The overview of joint fault types.

| Label | Failure State | Settings of General Force | |
| | | Y-Force | Z-Force |
|---|---|---|---|
| Norm | Normal | 0 | 0 |
| JF1 | IF | IF Excitation_C1 | IF Excitation_C2 |
| JF2 | OF in Z-axis | 0 | OF Excitation |
| JF3 | OF in Y-axis | OF Excitation | 0 |
| JF4 | IF + OF | OF Excitation + IF Excitation_C1 | IF Excitation_C2 |
| JF5 | OF + OF | OF Excitation | OF Excitation |

### 4.2.2. Applications of DCNNs

DCNN is suitable for learning hierarchical representation from images. It can usually extract general image features such as edges and curves from lower-level layers, which is suitable for most image classification tasks. Additionally, higher-level images tend to learn more abstract representations, which is suitable for a few special tasks. Therefore, DCNNs can be conducted to transfer learning (TL) by transferring weights in the lower level and fine-tuning the weights of the higher hidden layers. In this way, some powerful DCNNs for image recognition can be applied to other fields, such as the fault diagnosis problem in this work. In the previous section, the feature data representing the running states has been transformed to the same data form as images by CWT, so the relationship between image recognition and fault diagnosis is established, and TL should be applied under this precondition.

There are multiple DCNNs are available to be called from the Keras library, whose advanced performance has been verified on a large-scale image recognition dataset named "ImageNet", and their weights trained according to it are open-source and available to be loaded. Hence, the researchers can use them to solve new problems based on these previous achievements. The DCNNs used in this work include Xception [29], MobileNet [30], DenseNet121 [31], Resnet50 [32], InceptionV3 [33], and InceptionResNetV2 [34]. These DCNNs are loaded by removing top layers and replacing them with global average pooling (GAP), and they are also followed by fully connected layers with softmax regression as classifiers. Table 7 shows the relevant parameter settings for training DCNNs, and their performance achieved on the SF fault dataset is shown in Table 8. All of them achieve high precisions over 99% both on the training set and testing set, and DenseNet121 is selected as the DCNN model used in comparative experiments by both taking accuracy and model size into account.

**Table 7.** The relevant parameter settings for DCNN network.

| Parameter Item | Value |
|---|---|
| Learning rate | 0.001 |
| Loss function | Cross-entropy loss |
| Optimizer | Adam |
| Epoch | 15 |
| Batch size | 20 |
| Validation ratio | 0.1 |

**Table 8.** The recognition accuracies of DCNNs applied on SF dataset.

| Model | Training Accuracy | Training Accuracy | Parameters |
|---|---|---|---|
| Xception | 1.0 | 1.0 | 22,190,480 |
| MobileNet | 0.9911 | 0.9983 | 3,538,984 |
| DenseNet121 | 0.9967 | 0.9983 | 8,062,504 |
| Resnet50 | 0.9967 | 1.0 | 25,636,712 |
| InceptionV3 | 0.9959 | 0.9967 | 23,851,784 |
| InceptionResNetV2 | 0.9937 | 0.9967 | 55,873,736 |

### 4.2.3. Diagnosis Results of Comparative Experiments

Table 9 illustrates the precisions, recalls, and F1-scores obtained by DCNN both on SF and MF datasets, and the different motions of multi-joint robots are also separately treated in this stage. Here, the SF dataset is acquired by setting different excitations of joint 2, and the MF dataset is acquired based on the SF dataset by additionally applying fault excitations on joint 3. The cases of other joints are available to be observed in a similar way. It can be seen that DCNN has outstanding performance in most cases, even though the results of MF are slightly inferior to SF. In addition, the difference in results obtained under motion 1 and motion 2 is negligible, which further verifies that different motions have little influence on diagnosis results.

**Table 9.** The recognition results of DCNN on SF and MF dataset.

| Case | Label | Motion 1 | | | Motion 2 | | |
|------|-------|-----------|--------|----------|-----------|--------|----------|
| | | Precision | Recall | F1-Score | Precision | Recall | F1-Score |
| SF | Norm | 1.0000 | 1.0000 | 1.0000 | 0.9901 | 1.0000 | 0.9950 |
| | JF1 | 0.9900 | 0.9900 | 0.9900 | 1.0000 | 1.0000 | 1.0000 |
| | JF2 | 1.0000 | 0.9900 | 0.9950 | 1.0000 | 0.9800 | 0.9899 |
| | JF3 | 0.9900 | 1.0000 | 0.9950 | 1.0000 | 0.8900 | 0.9418 |
| | JF4 | 1.0000 | 1.0000 | 1.0000 | 1.0000 | 1.0000 | 1.0000 |
| | JF5 | 1.0000 | 1.0000 | 1.0000 | 0.8929 | 1.0000 | 0.9434 |
| MF | Norm | 0.9588 | 1.0000 | 0.9789 | 1.0000 | 0.9892 | 0.9946 |
| | JF1 | 1.0000 | 1.0000 | 1.0000 | 0.9688 | 0.9300 | 0.9490 |
| | JF2 | 1.0000 | 1.0000 | 1.0000 | 0.9891 | 0.9579 | 0.9733 |
| | JF3 | 0.9787 | 0.9583 | 0.9684 | 1.0000 | 0.9479 | 0.9733 |
| | JF4 | 1.0000 | 0.9811 | 0.9905 | 1.0000 | 0.9151 | 0.9557 |
| | JF5 | 1.0000 | 1.0000 | 1.0000 | 0.8333 | 1.0000 | 0.9091 |

DCNN is also compared to other algorithms, the results of which are shown in Figure 12. Among these algorithms, BPNN, SVM, and NBC follow the same principle as described in the previous section, and the comparative algorithm 'CNN' refers to a conventional CNN architecture that differs from the DCNN implemented in this work in that it only has two convolutional layers and uses one-dimensional convolution kernels. DCNN has the highest accuracies of each fault state, and other algorithms have errors under certain conditions. Most algorithms perform better on SF than MF, but in a few cases, the results may differ. This phenomenon is most likely caused by the instability of algorithm performance and the randomness of data. Furthermore, the difference in accuracies on SF and MF in motion 2 appears to be greater than in motion 1, which could be due to different recognition difficulties in different motion trials. However, anyway, DCNN has been validated as a suitable algorithm for recognizing the various fault types of rolling bearings.

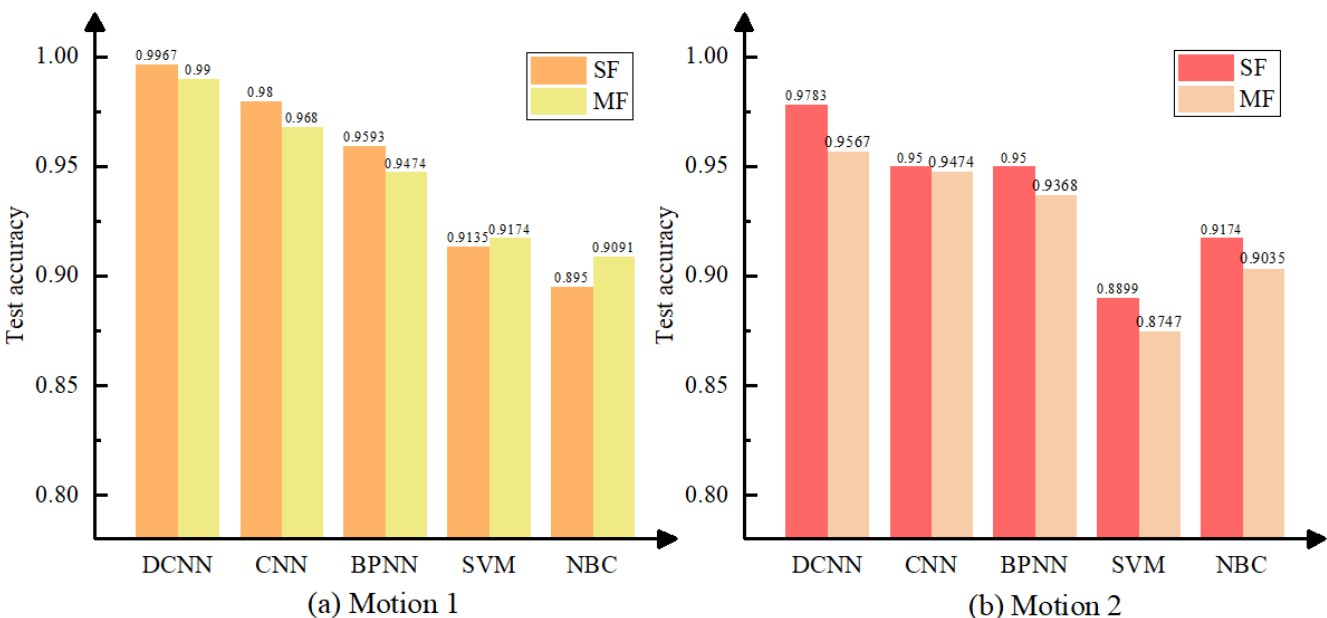

**Figure 12.** Comparisons of recognition accuracies in the second stage. (**a**) Motion 1; (**b**) Motion 2.

### 5. Conclusions

We conducted a complete procedure in this work, from dynamics modeling to fault diagnosis, for potential failures likely to exist in the rotating machinery of robot joints, and several valuable conclusions were obtained through this research process. First of all, the proposed simulation method is validated as feasible to simulate complex and diverse fault situations of the multi-joint robot due to the successful acquisition of vibration signals representing different failure features. Second, the LSTM is certified compatible with high accuracy in recognizing multi-channel vibration signals in diagnosis tasks such as locating faulty joints. Third, when time-frequency imaging is used, the DCNNs, which perform excellently in image recognition, can also be used to diagnose faults in rotating machinery. As a result, our proposed method for dynamics modeling and fault diagnosis is a feasible way to develop programs that recognize various types of failure in a mechanical system.

However, all our work is based on data acquired in a virtual simulation environment that differs from reality and does not take into account the effects of numerous external disturbance factors. It may cause performance reduction of our trained algorithms when they are directly used in reality. Therefore, some transfer learning ideas, such as domain adaptation, are probably necessary for future works to compensate for the difference between simulation and reality, which requires combining data collected from the real-world scene.

**Author Contributions:** Conceptualization, W.Z. and T.Z.; methodology, W.Z. and T.Z.; software, W.Z.; validation, W.Z.; formal analysis, W.Z., T.Z. and Y.P.; investigation, W.Z.; resources, W.Z., T.Z. and G.C.; data curation, W.Z. and T.Z.; writing—original draft preparation, W.Z.; writing—review and editing, W.Z. and T.Z.; visualization, W.Z.; supervision, T.Z. and Y.P.; project administration, T.Z.; funding acquisition, T.Z. and G.C. All authors have read and agreed to the published version of the manuscript.

**Funding:** This work was supported by the National Natural Science Foundation of China (NSFC) under Grant 11702168.

**Data Availability Statement:** Data presented in this study are available in this article.

**Conflicts of Interest:** The authors declare no conflict of interest.

### Abbreviations

The following abbreviations are used in this manuscript:

| SVM | Support Vector Machine |
| --- | --- |
| DL | Deep Learning |
| DBN | Deep Belief Network |
| DCNN | Deep Convolutional Neural Network |
| CNN | Convolutional Neural Network |
| LSTM | Long Short-Term Memory |
| BPNN | Back Propagation Neural Network |
| NBC | Naive Bayes Classifier |
| D-H | Denavit-Hartenberg |
| OF | Outer Race Fault |
| IF | Inner Race Fault |
| SF | Single-Joint Failure |
| MF | Multi-Joint Failure |
| JF | Joint Failure |
| RNN | Recurrent Neural Network |
| WPT | Wavelet Packet Transform |
| CWT | Continuous Wavelet Transform |

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
