# Peer review of "Dynamics Analysis and Deep Learning-Based Fault Diagnosis of Defective Rolling Element Bearing on the Multi-Joint Robot"

_machines, doi:10.3390/machines10121215_

Round 1

Reviewer 1 Report

The manuscrpt presents an pproach based on LSTM and CNN to etimate the location and the type of faults in the bearings of multi-dof robot arms.

Even though the topic might be of inerest, given the need to prevent mulfunctioning in robotic systems, I believe the work still needs major improvements.

First of all, the method is only applied ona simulation environment with synthetic dataset, appropriately made up by the authors. I think this greatly limitatates the impact of the work. 

From the experimental results, it seems like the authors manage to predict the location and type of fault but only on a specific robot motion, that only lasts 10 seconds. What if a more general sets of motions is used? How about the application in a real scenario?

Additionally, it is not clear how the anomalous states are "added" to the healthy states. Is it just a sum of values?

Whywould the authors need two networks? Why not using only one or combining the two somehow, since you are already extracting features from your data.

From a dontent perspective, the desdriptions of the robot kinematics, dynamics, and of the eural networks could be largely reduced by removing the mathematical derivations. I do not think they add value to the work and they are also generally well known.

Author Response

Thanks for advice! We have made a large adjustment of our article, and the answers to proposed questions can be seen in the attachment.

Reviewer 2 Report

In this paper, the authors conducted a complete procedure from dynamics modeling to fault diagnosis for the potential failures probably existing in the rotating machinery of robot  joints. 

1. The authors should give more detail descriptions of the LSTM structure used in this paper.

2. The DCNNs used in second stage are all based on existing work. Are there any theoretical innovations of your own work? Please give a remark here. 

3. According to the first stage designboth single joint failure and multiple joint failure cases should be described in second stage.

4. The details of model training, such as loss function, optimization algorithm and learning rate should be discussed. 

5. Analytic model based approach is an important research issue which is widely applied on fault diagnosis of rotary machines. The authors should supplement some results on this aspect, for example the following references had given significant design results:

[1] Incipient winding fault detection and diagnosis for squirrel-cage induction motors equipped
on CRH trains
. ISA Transactions, 2020, 99: 488~495.

Author Response

Thanks for advice! We have adjusted our article according to advice. The answer can be seen in attachment

Round 2

Reviewer 1 Report

I thank the authors for their effort in the replies.

I acknowledge the difficulties in terms of designing experiments close to the real scenarios and in computational power needed. However, the major limitation I see in this work is still in its applicability.

In reponse to question 2 about the specific motion types used, the authors state that they can train a new model for each new motion and that it is not an issue because robots in manufacturing generally have pre-defined motions.

I believe this is a major limitation because unexpected motions may actually occur. More and more we are seeing research focusing on programming robots as least as possible and make them learn tasks. Additionally, even in structured environments, often times robots need t correct their motions by using sensor fedback, and this might already lead to a different motion than the pre-defined one.

Author Response

Thanks for your question! We have answered it in the attachment, please check and approve. In conclusion, we think this problem doesn't have big impacts so there are no additional changes in the content of new revised manuscript. The changes of the article is mainly about polishing language.

Reviewer 2 Report

In a general way most of my comments were answered by the authors. My overall opinion about this paper is good. Please further polish English, in particular, the Abstract, Introduction and Conclusion section.

Author Response

Thanks for your recognition! We will pay more attenstions about the polishing of language, and improve the expressions about these mentioned parts. And in the newly revised manuscript, the language changes of Abstract, Introduction and Conclusion section are visible.